# Dynamical Signatures of Learning in Recurrent Networks

## Abstract

Recurrent neural networks (RNNs) are powerful computational tools that operate best near the edge of chaos, where small perturbations in neuronal firing are transmitted between neurons with minimal amplification or loss. In this article, we depart from the observation that both stimulus and noise can be seen as perturbations to the intrinsic dynamics of a recurrent network, however stimulus information must be reliably preserved, while noise must be discarded. First, we show that self-organizing recurrent networks (SORNs) that learn the spatio-temporal structure of their inputs, increase their recurrent memory by preferentially propagating the relevant stimulus-specific structured signals, while becoming more robust to random perturbation. We find that the computational advantages gained through self-supervised learning are accompanied by a shift from critical to ordered dynamics, and that this dynamical shift varies with the structure of the stimulus. Next, we show that SORNs with subcritical dynamics can outperform their random RNNs counterparts with critical dynamics, on a range of tasks, including a temporal MNIST and a sequential shape-rotation task. Interestingly, when a shape is rotated, both the invariant (shape) and the variant (motion direction) aspects of the stimulus sequence are improved through learning in the subcritical SORNs. We propose that the shift in criticality is a signature of specialization and we expect it to be found in all cases in which general-purpose recurrent networks acquire self-correcting properties by internalizing the statistical structure of their inputs.

## 1 Introduction

Randomly-connected recurrent neural networks (RNNs), often referred to as 'reservoir networks' Maass et al. (2002); Jaeger & Haas (2004), are powerful computational tools that mimic the brain's mastery for sequential processing Buonomano & Maass (2009), while achieving impressive performance across a large variety of applications. As dynamical systems, RNNs exhibit a fading memory of recent inputs which, in the limit, can be mapped through a simple readout to approximate any target function Maass & Markram (2004). In their seminal work, Bertschinger and Natschläger established a relationship between the computational power of recurrent networks and their dynamical properties Bertschinger & Natschläger (2004). Specifically, they showed that randomly connected networks perform best when they operate at the critical edge between stability and chaos.

While random RNNs with critical dynamics exhibit a strong memory of recent inputs, for any *arbitrary* input sequences, they differ substantially from real cortical networks which are remarkably structured and specialized. *Why did our brains evolve so as to contain so many specialized parts?* ask Minsky and Papert in Perceptrons Minsky & Papert (1969). *In order for a machine to learn to recognize or perform X, be it a pattern or a process, that machine must in one sense or another learn to represent or embody X.* Thus, it is sensible to believe that a good model for, for example, visual processing is one that captures internally the common contingencies of features present in the natural visual environment. It is currently unknown how such learning or specialization impacts the dynamics of a RNN.

In this paper we asked what is the relationship between network dynamics and performance, when RNNs learn the spatio-temporal structure of their inputs. We contrast random RNNs, tuned close to criticality, to self-organizing recurrent networks (SORNs) that learn via unsupervised biologically-

plausible plasticity mechanisms the sequential structure of their inputs Lazar et al. (2009); Hartmann et al. (2015).

## 2 NETWORK MODEL AND DYNAMICS

### 2.1 RECURRENT NEURAL NETWORK

The network model consists of 80% excitatory units ($N^E$) and 20% inhibitory units ($N^I$). $W^{IE}$, $W^{EI}$ and $W^{II}$ are dense connectivity matrices randomly drawn from the interval [0,1] and normalized so that the incoming connections to each neuron sum up to a constant ($\sum_j W_{ij} = 1$). The connections between excitatory units $W^{EE}$ are random and sparse ($p^{EE}$ is the probability of a connection), with no self-recurrence.

For each discrete time step $t$, the network state is given by the two binary vectors $\mathbf{x}(t) \in \{0, 1\}^{N^E}$, and $\mathbf{y}(t) \in \{0, 1\}^{N^I}$, representing activity of the excitatory and inhibitory units, respectively. The network evolves using the following update functions:

$$\mathbf{x}(t+1) = \boldsymbol{\Theta}(\mathbf{W}^{EE}(t)\mathbf{x}(t) - \mathbf{W}^{EI}\mathbf{y}(t) + \mathbf{U}(t)) - \mathbf{T}^E(t)) \tag{1}$$

$$\mathbf{y}(t+1) = \boldsymbol{\Theta}(\mathbf{W}^{IE}(t)\mathbf{x}(t) - \mathbf{W}^{II}\mathbf{y}(t) - \mathbf{T}^I) \tag{2}$$

The Heaviside step function $\Theta$ constrains the network activation at time $t$ to a binary representation: a neuron fires if the total drive it receives is greater than its threshold. The threshold values for excitatory ($T^E$) and inhibitory units ($T^I$) are drawn from a uniform distribution in the interval [0, $T^E_{max}$] and [0, $T^I_{max}$]. The initial values for excitatory weights ($W^{EE}$) and thresholds ($T^E$) stay constant for RNNs. These parameters have been chosen such that the dynamics of RNNs is close to criticality and their memory performance is high. SORNs change their weights and thresholds during stimulation in an unsupervised fashion following the rules described in Section 2.3.

The input U(t) varies as a function of time and goes to a subset $N^U$ of excitatory units.

### 2.2 PERTURBATION ANALYSIS

To define the chaotic and ordered phase of a RNN we use the perturbation approach proposed by Derrida and Pomeau for autonomous systems Derrida & Pomeau (1986) and employed by Bertschinger and Natschläger for RNNs Bertschinger & Natschläger (2004). At each timestep $t$, we flip the activity of one of the reservoir units (excitatory, non-input), giving us a perturbed state vector $x'(t)$ with a Hamming distance of 1 from the original network $x(t)$. We map these two states $x(t)$ and $x'(t)$ to their corresponding successor states $x(t + 1)$ and $x'(t + 1)$, using the same weights and thresholds, and we quantify the change in the Hamming distance. Distances that amplify are a signature of chaos, whereas distances that decrease are a signature of order.

### 2.3 SELF-ORGANIZING RECURRENT NETWORKS

For learning, we utilize a simple model of spike-timing dependent plasticity STDP that strengthens (weakens) the synaptic weight $W^{EE}$ by a fixed amount $\eta = 0.001$ whenever unit $i$ is active in the time step following (preceding) activation of unit $j$.

$$\Delta\mathbf{W}^{EE}_{ij}(t) = \eta_{STDP}(\mathbf{x}_i(t)\mathbf{x}_j(t-1) - \mathbf{x}_i(t-1)\mathbf{x}_j(t)). \tag{3}$$

In addition, synaptic normalization is used to proportionally adjust the values of incoming connections to a neuron so that they sum up to a constant value.

$$\Delta\mathbf{W}^{EE}_{ij}(t) = \mathbf{W}^{EE}_{ij}(t)/\sum_j \mathbf{W}^{EE}_{ij}(t). \tag{4}$$

To stabilize learning, we utilize a homeostatic intrinsic plasticity (IP) rule that spreads the activity evenly across units, by modulating their excitability using a learning rate $\eta_{ip} = 0.001$. At each

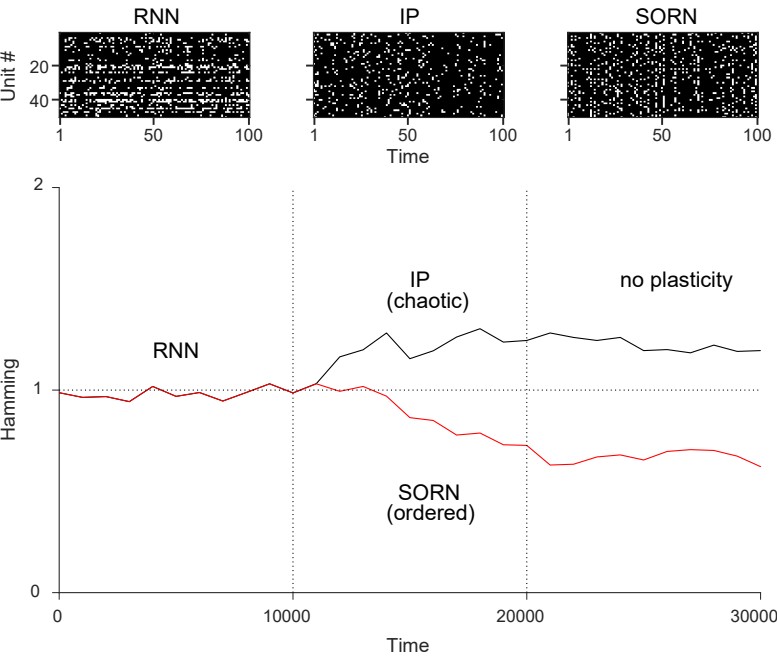

Figure 1: **Impact of self-organized learning on criticality** Recurrent networks of randomly connected excitatory units exhibit ordered, critical and chaotic dynamics. In the upper row, examples of the temporal evolution of the network state for a recurrent network with critical dynamics before and after the introduction of local plasticity rules. IP makes the dynamics of the network more chaotic (Hamming distance > 1), while SORN (STDP+IP) makes the dynamics more ordered (Hamming distance < 1). When plasticity is turned off, these recently acquired dynamical properties are preserved.

timestep, an active unit increases its threshold, while an inactive unit lowers its threshold by a small amount, such that on average each excitatory neuron will fire with the target firing rate $\mu_{IP}$:

$$\Delta \mathbf{T}_i^E = \eta_{IP}(\mathbf{x}_i(t) - \mu_{IP}) \tag{5}$$

Learning is a source of stability for RNNs. We find that as SORNs learn the spatio-temporal structure of their inputs, the dynamical regime exhibited by the recurrent network becomes more ordered (illustration in Figure 1). In contrast, when used in isolation, IP is a source of variability for RNNs. As the activity is spread evenly across units, the dynamical regime of the IP-only networks becomes more chaotic (Figure 1).

The implementation of the model described above and the simulations presented in the following sections were performed in Matlab.

## 3    DYNAMICAL SIGNATURES OF LEARNING

In the following simulations, we select initial parameter settings that result in network dynamics close to criticality, which are known to result in a high memory performance. An example of network activity for a reservoir composed of $N^E = 100$ excitatory units, $N^I = 25$, $p^{EE} = 0.05$, $\mathbf{T}_{max}^E = 0.5$, $\mathbf{T}_{max}^I = 0.3$ is presented in Figure 2. In order to understand the roles played by the homeostatic and learning plasticity mechanisms, we introduce them separately. Each plasticity interval (10000 steps) is followed by a no plasticity interval (10000 steps). Network performance and criticality are compared in the absence of plasticity.

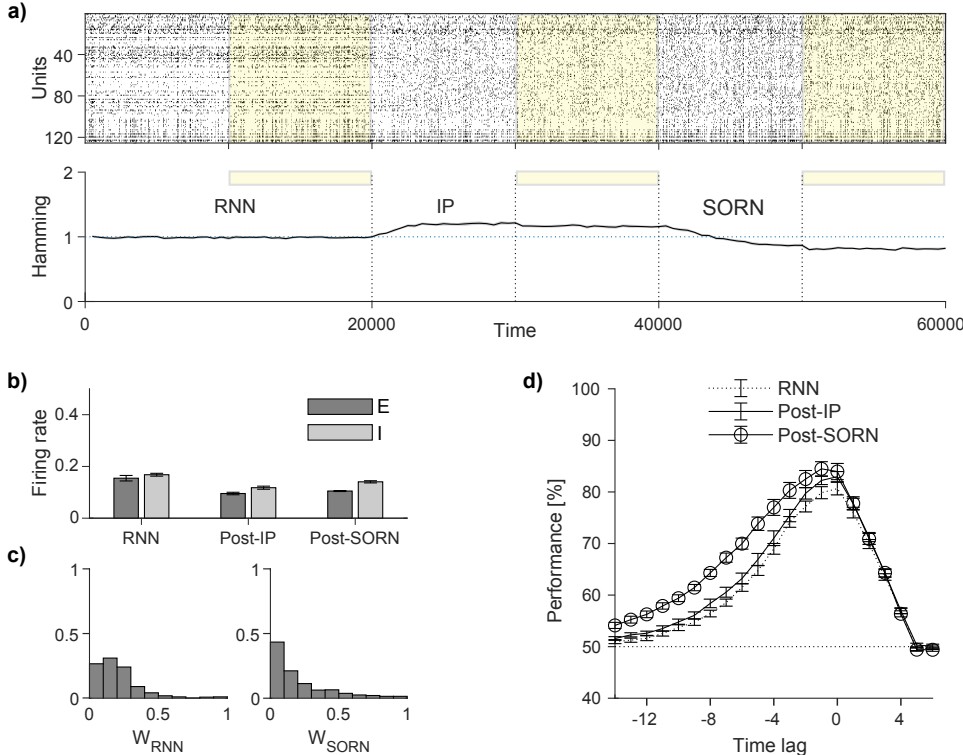

Figure 2: **(a)** Activity of a network composed of 125 units over 60000 time steps **(Top)** and corresponding perturbation analysis **(Bottom)**. Hamming distance is averaged across 500 consecutive time steps. The moments when IP and STDP+IP are introduced are marked by dotted lines. Networks with IP become more chaotic (Hamming distance>1). SORNs become more ordered (Hamming distance <1). In the absence of plasticity the networks retain their newly acquired dynamics. Shaded intervals are compared in (b) and (d). **(b)** IP-networks and SORNs have lower firing rates compared to random RNNs tuned to exhibit critical dynamics, because of their homeostatic IP. **(c)** Example of weight distributions before and after unsupervised learning with STDP. **(d)** Memory and prediction performance scores are compared between random RNNs, IP-networks and SORNs, after plasticity was turned off (shaded intervals in (a), 20 runs). SORNs have an increased memory for the experienced stimuli compared to their counterparts.

In a first learning task we randomly alternate two input sequences of length 5 ($a_1, a_2.., a_5; b_1, b_2.., b_5$). Each symbol in a sequence activates 2 input neurons, thus $N^U = 20$ units out of the total 100 excitatory units receive input. Using the perturbation analysis described in Section 2.2, we observe a shift in network dynamics towards a chaotic regime (Hamming>1) after IP is introduced (Figure 2a, average over 20 network runs). When STDP is introduced in addition, the network dynamics becomes ordered (Hamming<1).

Interestingly, while the firing rates of IP-networks and SORN-networks are similar (Figure 2b), due to the presence of IP, the dynamical regimes exhibited by these two types of networks are very different.

Through learning, SORNs change their synaptic weight distribution, with a subset of weights becoming stronger and a subset of weights becoming weaker (example run in Figure 2c).

To quantify the computational power of the different recurrent networks under comparison, a simple Bayesian readout was trained to differentiate between the two sequences based in the reservoir state (excitatory non-input units) of the network at the present time plus or minus a time lag. A *k*-fold validation procedure was applied, that split data into *k* partitions, with *k*-1 partitions used for training and 1 partition used for test. We found that after 10000 steps of unsupervised plasticity, SORNs learned the predictable structure of these simple sequential inputs and significantly outperformed both RNNs and IP-networks on this task (Figure 2d, 20 network runs, $k = 4$ partitions).

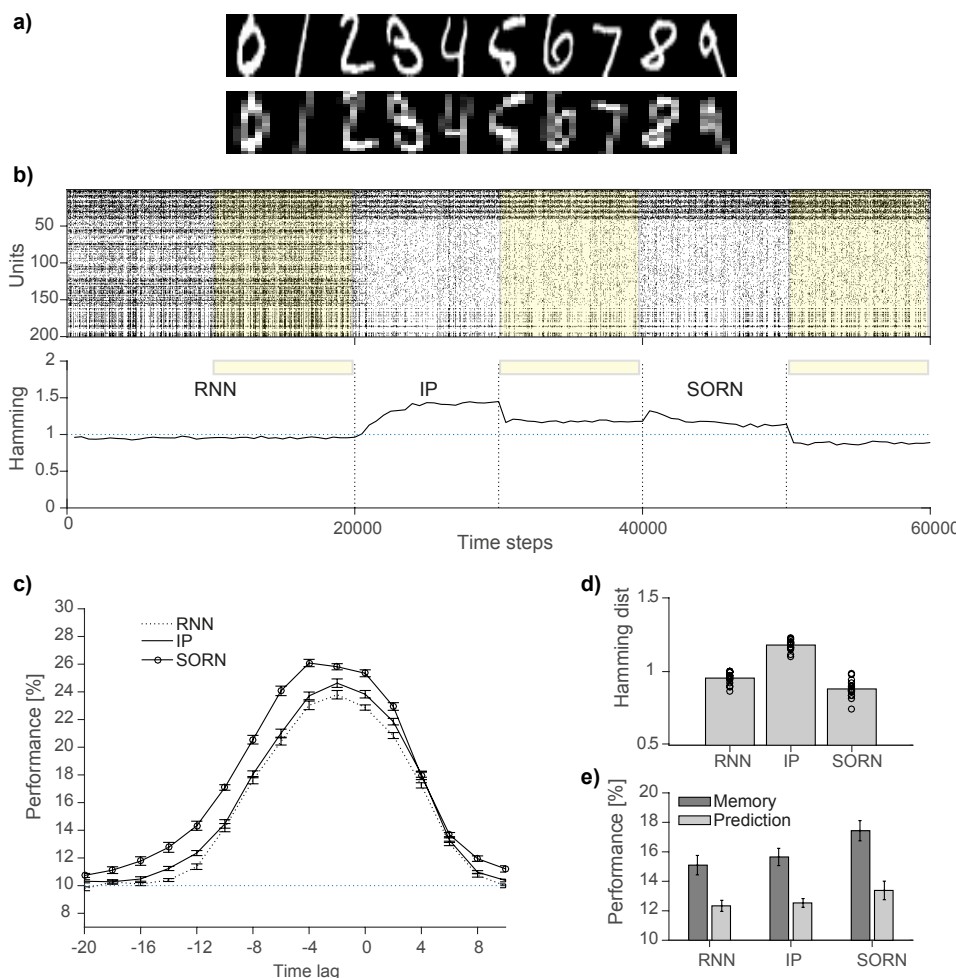

Figure 3: Impact of learning a temporal-MNIST on network dynamics. **(a)** Example stimuli from the original MNIST dataset and downsampled versions received by the network. 6000 stimuli were downsampled to 10x10 pixels and sectioned from top to bottom into 10 consecutive chunks, i.e. each digit became a vector sequence of length 10. **(b)** Activity of RNN, IP-network and SORN for 10000 steps and 10000 steps post-plasticity. The comparison is made between non-plastic networks, marked by shaded areas. RNNs are tuned to be close to criticality, where their performance is known to be high. Networks with IP become chaotic (Hamming distance>1) and remain chaotic when IP is turned off. SORNs become ordered after learning (Hamming distance<1). **(c)** Impact of learning a large dataset (MNIST images) on network performance. RNNs have been tuned to be close to criticality, where their performance is known to be high, but are outperformed by SORNs. **(d)** The dynamics of SORNs becomes ordered after learning. **(e)** Performance of SORNs is significantly higher for both memory and prediction compared to IP-networks and random RNNs (average over timelags 1 to 10).

## 4 Self-Organized Learning of MNIST dataset

In a second task, a larger set of inputs was employed. 6000 independent examples of hand-written digits (10 digits, 600 examples per digit) from the MNIST dataset were downsampled as pictured in Figure 3 and sectioned from top to bottom into 10 consecutive chunks (each image became an input sequence of length 10). The resulting sequential input was 60000 time steps long and contained no overlapping data (each hand-written digit was presented exactly once).

We selected initial parameter settings that resulted in network dynamics close to criticality. Specifically, the random recurrent networks were composed of $N^E = 160$ excitatory, $N^I = 40$ inhibitory

units, with $p^{EE} = 0.03$, $\mathbf{T}^E_{max} = 0.4$ and $\mathbf{T}^I_{max} = 0.3$. An example of network activity and dynamics can be observed in Figure 3. As before, intrinsic and synaptic plasticity mechanisms were introduced separately and were followed by 10000 steps of no plasticity. Network performance and criticality were compared in the absence of plasticity (intervals marked in yellow).

To assess the memory performance of a recurrent network, simple Bayesian readouts were trained to classify the current input digit based on network activity at time $t$ plus or minus a time lag. Results were averaged over 20 network run. For each network run a 4-fold validation procedure was employed: through rotation, three quarters of data were asigned for training, one quarter for test and the results were averaged. We found that the performance of SORNs was significantly higher than that of IP-networks and random RNNs for both memory and prediction (Figure 3c and e).

The dynamics of networks after learning was ordered, Hamming distance scores for SORNs were significantly lower than the Hamming distance scores of random RNNs (t-test, $p < 0.0005$, Figure 3d) and significantly lower than 1 (t-test, $p < 4 \times 10^{-6}$).

## 5   Self-Organized Learning of Rotating Shapes

In a final task, we employed rotating shapes to differentiate between different aspects of sequential processing. Two shapes were rotated 90 degrees, either left or right, across seven individual image frames (Figure 4a). The network activity was split into individual trials, the rotating shapes alternated randomly across trials. Each trial consisted of 80 time-steps: 3 time-steps per image frame, 2 time-steps blank between frames, and 40 time-steps of activity after the stimulus presentation, which allowed us to quantify the recurrent memory traces. Example trials for the three network types are shown in Figure 4b, in all three cases the networks respond strongly to an example stimulus sequence and produce recurrent activity after the stimulus is extinguished.

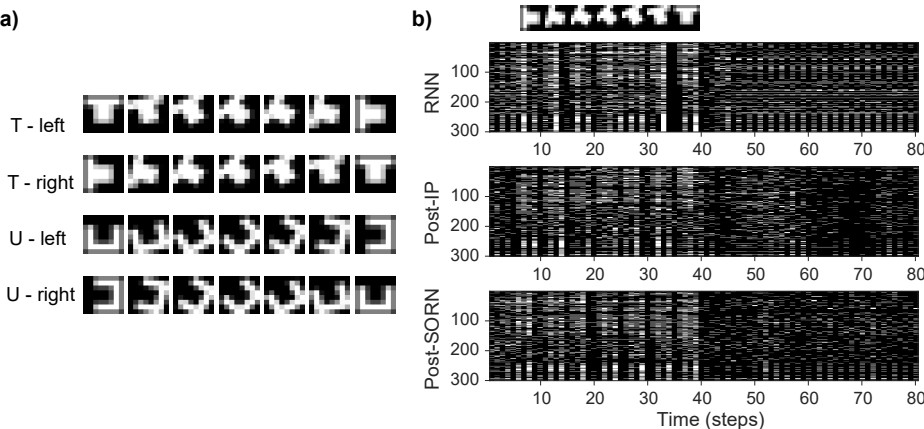

Figure 4: Shape-rotation task. **(a)** Left and right rotating shapes. Each sequence is composed of 7 frames, each frame is 8x8 pixels. **(b)** Example single-trial network responses to a sequence presentation for the three types of networks.

The network activity of a reservoir with $N^E = 240$ excitatory units, $N^I = 60$, $p^{EE} = 0.02$, $\mathbf{T}^E_{max} = 0.5$, $\mathbf{T}^I_{max} = 0.3$ is shown in Figure 5a. As in the previous tasks, the Hamming distance increased for IP-networks and decreased for SORNs (Figure 5a bottom, 480 trials, 38400 timesteps).

We employed Bayesian classifiers to decode *stimulus shape*, irrespective of motion direction (Figure 5b) and *motion direction*, irrespective of stimulus shape (Figure 5c). In the first task, decoding of stimulus shape, maximum peak performance was reached for all three network types during stimulus presentation, even though the Bayesian classifiers were trained exclusively on reservoir units (input units were excluded). SORNs exhibited higher performance after the stimulus offset, compared to the other networks, with decoding performance being at $\approx 70\%$ at the end of the trial (Figure 5b).

The second task, decoding of motion direction, was more difficult, since it required an integration of information over time. We found that a strong memory trace of information about rotation direction was available in SORNs, $> 65\%$, at the end of the trial (Figure5c).

We conclude that the shift towards more orderly dynamics can be associated with enhancements in both variant and invariant aspects of sequential processing in self-organizing recurrent networks.

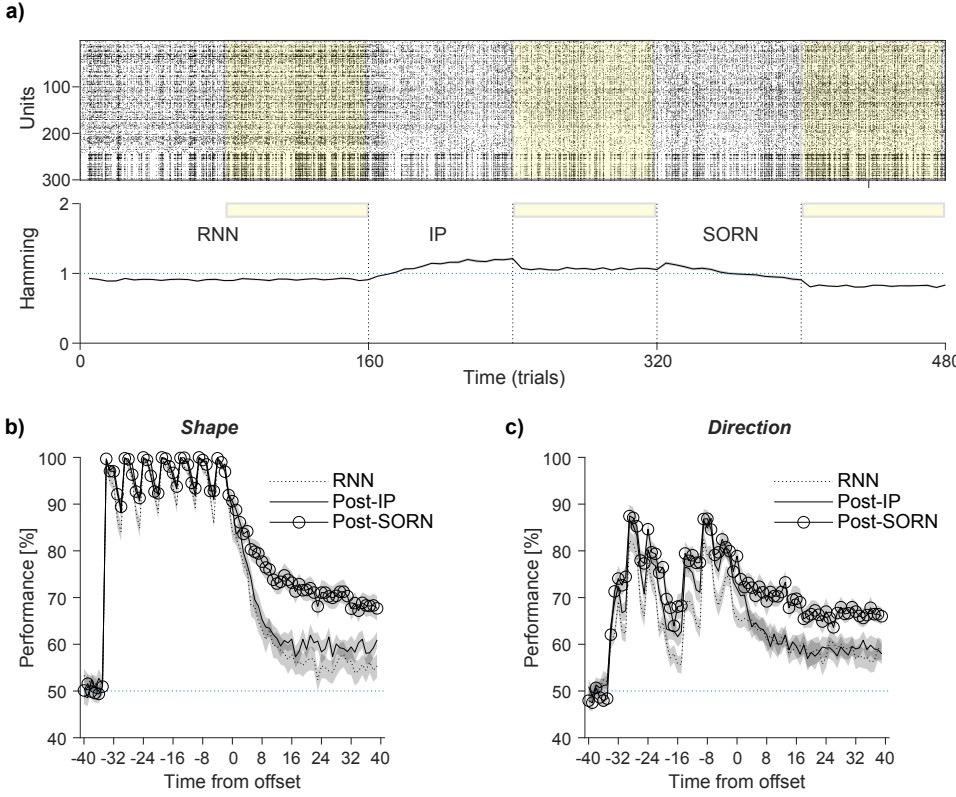

Figure 5: **(a)** Activity of a network composed of 300 units and corresponding perturbation analysis during the shape-rotation task over 480 trials (38400 time steps). Random RNNs become more chaotic after IP and more ordered after self-organization. Comparison of the memory performance of the three network types (yellow markings) in decoding the *shape* **(b)** and *rotation direction* **(c)** of the object. Negative times correspond to the stimulus presentation, 0 marks the stimulus offset. After learning, SORNs exhibit a much stronger memory trace for both shape and direction compared to the other network types.

## 6    DISCUSSION

*It makes no sense to seek the best network architecture or learning procedure because it makes no sense to say that any network is efficient by itself: that makes sense only in the context of some class of problems to be solved.* Minsky & Papert (1969)

We compared the dynamical properties of general-purpose recurrent networks to those of *specialized* recurrent networks that have learned the sequential structure of their inputs. We showed that local plasticity rules for learning and homeostasis can have opposing influences on network dynamics. Most importantly, we found that recurrent networks that learn the spatio-temporal structure of their inputs exhibit enhanced computational abilities and concomitantly shift their dynamics towards the ordered regime. The increased robustness to perturbation exhibited after learning provides us with hints regarding the dynamical principles that enable biological systems to perform sophisticated information processing.

Early theories of visual perception have suggested that the brain interprets its input signals on the basis of an internal model of the visual world Von Helmholtz (1867). Such a model can be learned through experience because the visual world is immensely rich, yet exquisitely structured, both across space and time. The ideal model, as it turns out, is not a general random machine, but a highly structured network that can accommodate a large number of statistical contingencies, as required for the interpretation of ever changing sensory input patterns. Here we have shown, in simple recurrent models, that the signatures of network specialization can be picked up from the network's dynamics. Future experimental work on cortical networks could investigate whether such a correspondence exists in real brain networks and to what extent learning increases their robustness to perturbation, or their ability to process noisy stimuli.

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

## A   Appendix

### Model parameters

| Figure | Units | $N^E$ | $N^I$ | $N^U$ | $p^{EE}$ | $p_{IE}$ | $p_{EI}$ | $p_{II}$ | $TE_{max}$ | $TI_{max}$ | $rate_E$ | Time |
|--------|-------|-------|-------|-------|----------|----------|----------|----------|------------|------------|----------|------|
| 1 | 200 | 160 | 40 | 60 | 0.05 | 1 | 0.5 | 1 | 0.4 | 0.3 | 0.1 | 30000 |
| 2 | 125 | 100 | 25 | 20 | 0.05 | 1 | 0.5 | 1 | 0.5 | 0.3 | 0.1 | 60000 |
| 3 | 200 | 160 | 40 | 40 | 0.03 | 1 | 0.5 | 1 | 0.4 | 0.3 | 0.1 | 60000 |
| 5 | 300 | 240 | 60 | 64 | 0.02 | 1 | 0.5 | 1 | 0.5 | 0.3 | 0.1 | 38400 |

### Dependance of criticality and performance on task difficulty

We use a learning task in which we randomly alternate two input sequences of length $n + 2$ ($a, x.., x, b$; $c, x.., x, d$), where input $x$ is common to both sequences and repeated $n$ times. The two input sequences alternate randomly during stimulation and are separated by up to 5 blanks. A linear readout is trained to separate between the two sequences (blanks removed). Each symbol in a sequence activates 5 input neurons, thus a total of 25 units receive input. The interesting aspect of this task is that the the length of the input sequences can be varied without changing the ratio of input units to reservoir units. We simulate 20 networks for each $n = \{4, 8, 12, 16, 20, 24\}$. The longer sequences require a longer recurrent memory trace for correct classification. We find that the fading memory of SORNs improves through unsupervised learning over that of RNNs for both short and long sequences. For very long sequences ($n = 24$, see Figure 1, right), the maintanance of a long memory trace comes with a cost for decoding recent and current stimuli.

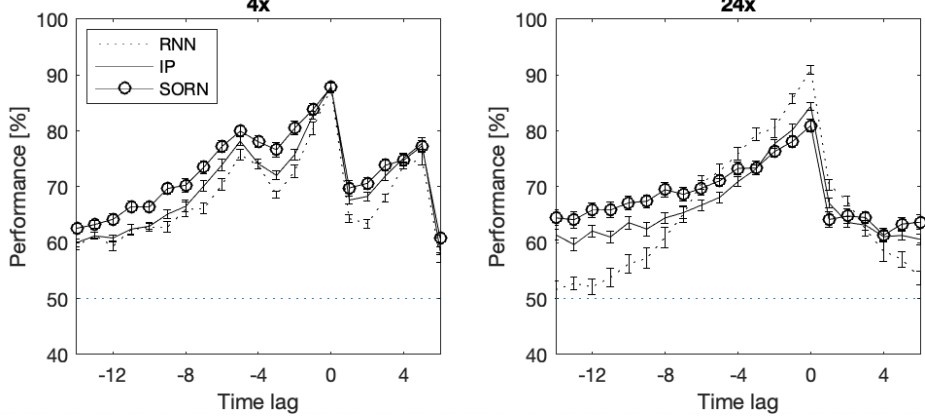

Figure 6: Contrast of learning short ($n = 4$, left panel) vs long sequences ($n = 24$, right panel). RNNs with N=125 units, $p^{EE} = 0.05$, $\mathbf{T}^E_{max} = 0.4$, $\mathbf{T}^I_{max} = 0.3$ are shaped by IP or IP+STDP (SORN). SORNs maintain a longer input memory trace compared to IP-networks and RNNs. Error bars indicate s.e.m.

We compared the criticality values, estimated via a perturbation analysis, for the three types of reservoirs considered here, to their memory performance on the task above (time lag = -15:-5). The firing rates of IP-networks and SORNs are nearly identical, due to the presence of IP with a set target rate for all units, however the dynamical regimes exhibited by these two types of recurrent networks are quite different. IP-networks exhibit chaotic dynamics while SORNs exhibit ordered dynamics for all values of $n$ (Figure 2, left panel). While SORNs exhibit the most orderly dynamics for smaller $n$, they outperform both IP-networks and RNNs on the memory task by similar amounts for all values of $n$. This interesting result suggests that, in SORNs, a perturbation at a start of an input sequence is more likely to be corrected than a perturbation at the end of an input sequence. This type of stimulus dependance is less pronounced for IP-networks and RNNs. Prompted by this result we initiated an analysis that compares outcomes of perturbations of the reservoir's state with respect to their occurrence inside an input sequence, which we plan to include in the final manuscript.

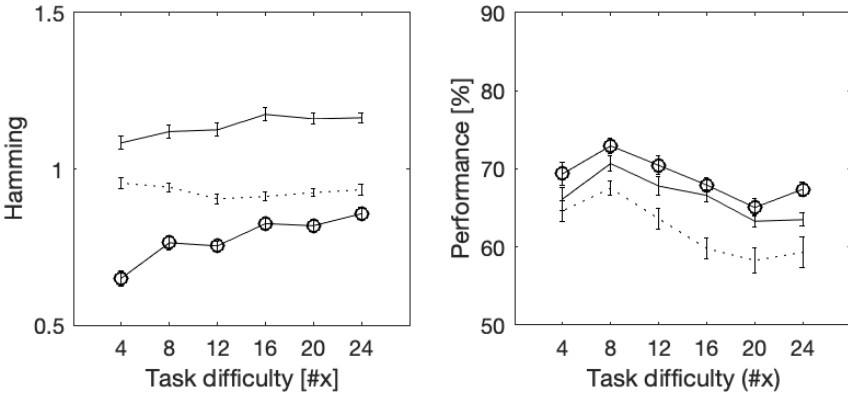

Figure 7: Hamming distance for RNNs, IP-networks and SORNs in the presence of inputs of varying length. SORNs outperform the other networks and exhibit ordered dynamics (Hamming distance <1) for all values of $n$. Most orderly dynamics if observed for smaller $n$. Error bars indicate s.e.m.

