# OpenReview forum: "Dynamical Signatures of Learning in Recurrent Networks"
_ICLR.cc/2023/Conference — Submitted to ICLR 2023_

### Official Review · Reviewer_vp6e · 2022-10-24

**Confidence:** 5
**Correctness:** 3
**Technical Novelty And Significance:** 1
**Empirical Novelty And Significance:** 2
**Recommendation:** 3

**Clarity, Quality, Novelty And Reproducibility:**

The text is overall easy to read, although I found a few of the technical descriptions not precise enough and subject to interpretation. In particular, i was unclear on exactly how the readout works for each of the tasks (bayesian decoder is a little too vague). I was unsure in places if results reported were on the training or test set. For the first task, it seemed that nothing except for externally defined semantics differentiated elements from sequence a and b, since the temporal structure in the input space was purely periodic, i did not understand the logic of decoding a vs b at large lags.
Unclear if the code for the simulations was released with the manuscript, if so that would alleviate some of the clarity/reproducibility concerns.

**Strength And Weaknesses:**

+ Strength: SORNs involve purely local unsupervised updates; that endows them with some computational advantages relative to static reservoirs with randomly initialized parameters by exploiting statistical regularities in the inputs.
- Weakness: the choice of model seems out of touch with the field, which invariably uses differentiable nonlinearities for the units, and which have lent itself to perhaps more interesting analytical investigation of the RNN dynamics after learning
- W: the tasks are somewhat contrived to involve toy sequential structure
- W: performance seems relatively poor overall
- W: relevance of the work seems restricted to a small niche of computational neuroscience which focuses on criticality and reservoir computing, unclear what implications if any should this have on the wider comp neuro or machine learning field.

**Summary Of The Paper:**

The paper compares a RNN with step nonlinearities with randomly initialized parameters to ones in which parameters are trained by simple local updates inspired by biology. Namely, connections are changed to reinforce sequential structure (STDP) while the neuron thresholds change so as to keep activities small (IP). These RNN variants are evaluated in terms of their criticality via numerical perturbations and in terms of the ability of the network representations to carry information about sequential inputs to the network (using a separately trained decoder).

**Summary Of The Review:**

Overall, a simple model analyzed with numerical means using methodology similar to past SORN work (Lazar, Triesch, etc), on a few toy tasks. The results show slight benefits from SORN learning over no learning (arguably, the minimal precondition for usefulness); I would not say that the mechanics of the process are much better understood as a result of the accompanying analysis and the relevance for the broader community seems minimal.

---

### Official Review · Reviewer_RfJy · 2022-10-24

**Confidence:** 4
**Correctness:** 2
**Technical Novelty And Significance:** 3
**Empirical Novelty And Significance:** 3
**Recommendation:** 5

**Clarity, Quality, Novelty And Reproducibility:**

The methods and experimental results are clear. The investigated link between learning and dynamics in RNNs is very novel.

**Strength And Weaknesses:**

$\textbf{Strengths}$:

Studying the connection between dynamical properties and learning in artificial neural networks (ANN) is novel, and can potentially build stronger link between the two common approaches of dynamical systems and deep learning in neuroscience. I found the approach presented in the paper very useful in that sense.

$\textbf{Weaknesses}$:

I have some reservations about the main claim of the paper: namely “the shift in criticality is a signature of specialization.” I am not convinced that the data presented in the paper supports this general claim and is not specific to the learning rule and training sequences used in the experiments. I elaborate below:

$\textbf{Questions}$:

1- All the results in this paper are produced with a learning rule that is a combination of a Hebbian plasticity and a homeostatic plasticity rule. If the main claim of the paper is that gaining specialization (regardless of the learning rule) leads to subcriticality in the dynamics of the RNN, I'd expect to see similar results with other learning rules, e.g. Oja’s rule, BCM rule, gradient descent rules, etc.

2- The change in the Hamming distance after learning is different between different tasks. For example, in Figure 2a after applying the learning rule, the Hamming distance immediately starts going down, but in Figure 3b (the MNIST data), applying the learning rule actually causes a jump in the Hamming distance which gradually goes down but doesn’t get to below one (the subcriticality) until learning is turned off. This seems to suggest that there is an interaction between the effect of learning and the sequence statistics. How do the authors explain this apparent discrepancy between the two experiments?

3- Although the effect of homeostatic plasticity on the dynamics is investigated separately (e.g Figure 1), the effect of STDP isn’t examined separately. Therefore, it is not clear whether the shift from criticality to subcriticality is due to the STDP rule or it is a combined effect of STDP and homeostatic plasticity. I suggest that the authors also include a comparison with STDP alone, or if it is not possible, please explain the reason.

4- Related to comment # 2, the interaction between the statistics of the input sequence and the learning rule on shifting the dynamics of RNNs is not clear. How would the dynamical regime change if the input data was white noise? In a more systematic exploration, you could gradually increase the temporal (and even spatial) correlations (e.g. using an auto-regression process) and evaluate the shift in the dynamics as a function of the temporal/spatial correlation in the input sequence.

5- In the introduction it is mentioned that “random RNNs with critical dynamics exhibit a strong memory of recent inputs for any arbitrary input sequences.” The results presented in the paper, however, show that learning improves the memory of the RNN for the training sequences. But, it is expected that it should decrease memory performance for other arbitrary sequences. It would be useful to explicitly show the underperformance of the RNN with other sequences compared to an RNN functioning at the edge of chaos. This would give a more balanced picture of the advantages and disadvantages of specialization. For example, in the context of continual learning (which might be beyond the scope of this paper), becoming too specialized might be a disadvantage for the neural network.

$\textbf{Minor comments}$:

1- In section 2.1, all inhibitory and excitatory synaptic weights are explained as being sampled from [0,1]. Is this only the magnitude of the synaptic connections, given that the inhibitory connections are expected to be sampled from a range of negative values.

2- How are memory and prediction performance measured separately for the plots in Figure 3e?



**Summary Of The Paper:**

In this paper, the authors compare the dynamical characteristics of recurrent neural networks before and after training. Specifically, the results presented in the paper suggest that training shifts RNNs toward subcriticality. The authors suggest that this subcriticality is a signature of specialization that can potentially be used to study specialization in biological neural networks.

**Summary Of The Review:**

I believe this paper can make an important contribution to both machine learning and neuroscience by providing a dynamics-based metric for studying specialization in artificial and biological neural networks. However, I believe that the paper could benefit from more thorough examination of different learning rules and a few more control experiments, which I explained in my comments. I am willing to improve my score if the authors address my questions especially ones regarding different learning rules.

---

### Official Review · Reviewer_CT5J · 2022-10-24

**Confidence:** 4
**Correctness:** 2
**Technical Novelty And Significance:** 2
**Empirical Novelty And Significance:** 2
**Recommendation:** 3

**Clarity, Quality, Novelty And Reproducibility:**

The paper is clearly written, but the core idea is not well explained and verified.

**Strength And Weaknesses:**

Strength:

The idea is somehow interesting and the written is clear.

Weaknesses:

1, It is better to give an illustration of the network model, especially for those readers without comp. neuro. background. For the pertubabtion analysis, you can also give a illustration at somewhere.

2, Many parts of the paper need to be explained more clearly. For instance, what does the input looks like when producing Figure 1?

3, All the results are only verified with simulations, and theoretical analysis of why the network work is missing, making the paper more like a technical report.

**Summary Of The Paper:**

The authors showed that self-organizing recurrent networks which learn the spatio-temporal structure of their inputs, increase their recurrent memory by preferentially propagating the relevant stimulus specific structure signal, while becoming more robust to random pertubation. They also showed that the SORN model with subcritical dynamics outperfrom random RNN counterparts with critical dynamics on a range of tasks.

**Summary Of The Review:**

This paper is lack of many details and contributes only in a limited way to the ICLR society.

---

### Official Review · Reviewer_B89g · 2022-10-26

**Confidence:** 4
**Correctness:** 3
**Technical Novelty And Significance:** 2
**Empirical Novelty And Significance:** 2
**Recommendation:** 3

**Clarity, Quality, Novelty And Reproducibility:**

The work is original to my knowledge.

The clarity of the manuscript can be improved quite significantly, see comments above. Moreover, the motivation and the specific issue being investigated is not made very clear. The experimental setup seems to be used without justification as well.

Based on the details in the paper, reproducing the experiments would be challenging.

**Strength And Weaknesses:**

## Strengths:

- The properties of SORNs shown are very interesting in terms of such a sub-critical network performing better than RNNs at EOC.
- Being able to get better performance than a standard reservoir on these tasks based on SORN training is nice.

## Weaknesses:

- A lot of the experimental setup seems very arbitrary, and there are no ablation studies to determine what's important. e.g. does IP always have to precede SORN?
- The results of the study are hard to generalize due to very specific choices of plasticity and network setup.
- Some parts of the experimental details are not clear.

## Other questions

- In Fig. 2, why is the performance above chance for both positive and negative time lags?
- How's the bayesian readout trained?
- More explanation needed for what variant and invariant aspects of processing are in Sec. 5.
- How does SORN compare with self-supervised training using other optimization methods?

**Summary Of The Paper:**

In this paper, authors study the performance and properties of SORNs that have been exposed to the training dataset in terms of its ability to differentiate between task relevant and noise inputs. They show that these SORNs have sub-critical dynamics and can outperform RNNs at the edge of criticality in a couple of benchmarks.


**Summary Of The Review:**

Overall, while the issue investigated and the results are interesting, the paper needs to improve on multiple fronts -- better and clearer writing, clearer statement of question being investigated, better justifications for various experiment details, more experiments to judge generality of results.

---

### Decision · Program_Chairs · 2023-01-20

**Decision:**

Reject

**Justification For Why Not Higher Score:**

This paper is limited in application and lacking in clarity. Coupled with the lack of rebuttal and the scores, there is no solid argument in favour of accepting this paper.

**Justification For Why Not Lower Score:**

N/A

**Metareview: Summary, Strengths And Weaknesses:**

This paper presents an analysis of self-organized recurrent neural networks. It shows that these networks enter a sub-critical dynamics regime that makes them better at some toy tasks examined relative to networks that are maintained at a regime of critical dynamics.

The reviewers were in agreement that this paper has some interesting analyses, but ultimately, delivers very little in the way of novel insights, makes claims that are not supported, and lacks broader relevance to the ICLR community. The authors did not respond to the reviewers comments, so the scores were unchanged, giving an average score of 3.5. Thus, there was a consensus that this paper should be rejected.

**Summary Of Ac-Reviewer Meeting:**

N/A